# Roflumilast in Severely Ill Patients with Chronic Obstructive Pulmonary Disease with Frequent Exacerbations: Risk of Pneumonia Hospitalization and Severe Exacerbations

**DOI:** 10.3390/jcm9051442

**Published:** 2020-05-12

**Authors:** Imane Achir Alispahic, Rikke Sørensen, Josefin Eklöf, Pradeesh Sivapalan, Anders Løkke, Niels Seersholm, Jakob Hedemark Vestergaard, Jens-Ulrik Stæhr Jensen

**Affiliations:** 1Department of Internal Medicine, Respiratory Medicine Section, Herlev and Gentofte Hospital, University Hospital of Copenhagen, 2900 Copenhagen, Denmark; imane.achir.01@regionh.dk (I.A.A.); josefin.viktoria.ekloef@regionh.dk (J.E.); pradeesh.sivapalan.02@regionh.dk (P.S.); seersholm@dadlnet.dk (N.S.); jakob.hedemark.vestergaard.03@regionh.dk (J.H.V.); 2Department of Cardiology, Rigshospitalet, University Hospital of Copenhagen, University of Copenhagen, 2100 Copenhagen, Denmark; rikke.soerensen@regionh.dk; 3Department of Internal Medicine, Zealand University Hospital, 4000 Roskilde, Denmark; 4Department of Medicine, Vejle, Hospital Little Belt, 7100 Vejle, Denmark; aloekke@gmail.com; 5PERSIMUNE, Department of Infectious Diseases, Rigshospitalet, 2100 Copenhagen, Denmark

**Keywords:** COPD, roflumilast, hospitalization-requiring pneumonia, mortality

## Abstract

Roflumilast is given as an add-on to inhalation medication in patients with chronic obstructive pulmonary disease (COPD) and chronic bronchitis. Animal experiments have documented deleterious effects of roflumilast in bacterial infections, but trials have not reported the risk of bacterial infections in patients. The objective of this study is to determine, among outpatients with severe COPD in a two-year follow-up period, the risk of hospitalization-requiring pneumonia, severe acute exacerbation in COPD (AECOPD-hosp), and death. Patients with COPD using roflumilast (roflumilast users) were compared to a propensity score-matched COPD control group not using roflumilast (non-roflumilast users). Roflumilast users had an increased 2-year risk of hospitalization-requiring pneumonia (HR 1.5, 95% CI 1.3 to 1.8, *p*-value < 0.0001) compared to controls, and of AECOPD-Hosp (hazard ratio(HR) 1.6, 95%, confidence interval (CI) 1.5 to 1.8, *p*-value < 0.0001) and. When adding an active comparator (theophylline) as a matching variable, the signal was largely unchanged. In conclusion, roflumilast was associated with an increased number of hospitalizations for pneumonia and for AECOPD. Since trials have not reported risks of bacterial complications and data regarding severe exacerbations in roflumilast users are sparse and diverging, these data are concerning. Trials focused on the risk of pneumonia, AECOPD, and other bacterial infections in roflumilast users are needed urgently.

## 1. Introduction

Chronic obstructive pulmonary disease (COPD) is one of the leading causes of death and debilitation worldwide [1,2,3]. Patients with COPD may experience exacerbations and even COPD-related hospitalizations despite treatment of triple therapy with long-acting muscarinic antagonists (LAMA), long-acting beta2-agonists (LABA), and inhaled corticosteroids (ICS). Drugs exist for adjunctive therapy in patients with severe COPD who are not stabilized on inhalation therapy. Among these last-resort options, roflumilast has been introduced during the last decade [4,5,6,7,8,9,10,11,12,13,14,15].

Roflumilast is an oral selective inhibitor of phosphodiesterase-4 (PDE4) acting in the same mechanism astheophylline, a nonselective phosphordiesterase inhibitor. The drug can contribute to reducing inflammation in the lungs [16] and is approved for the treatment of COPD patients with a post-bronchodilatoryFEV1 (Forced Expiratory Volume in 1 s) less than 50% of predicted, chronic bronchitis, and with a known history of exacerbations [17]. Roflumilast is used as an “add-on” therapy for patients with COPD, and is administered as a tablet of 500 µg once daily. Randomized trials [4,10,11,12,13,14,15] have demonstrated an improvement in FEV1 (pre- and post-bronchodilatory) as well as a decrease in moderate-severe COPD exacerbations [7,9,10,18]. Of special concern, animal experiments have shown deleterious effects of roflumilast in bacterial lung infectious with increased mortality and bacterial loads [19], and likewise concerning, roflumilast has been documented to reduce the release of chemokine cysteine-cysteine ligand (CCL)2, CCL3, CCL4, chemokine cysteine-X-cystein ligand (CXCL) 10 and TNF-α, which may severely compromise the host defense towards bacterial infections [20,21]. However, specific results regarding the risk of roflumilast on hospitalization-requiring pneumonias have not been reported in trial data, and data on severe (hospitalization-requiring) COPD exacerbations have been sparse and diverging [10]. Additionally, to our knowledge, long-term follow-up for outcomes in patients with severe COPD using roflumilast have not been investigated in larger, real-life cohorts.

The aim of the current study was to determine if roflumilast is associated with a lower risk of hospitalization-requiring pneumonia, severe (hospitalization-requiring) COPD exacerbations (AECOPD-Hosp), and all-cause mortality in outpatients with COPD during a 2-year follow-up period.

## 2. Experimental Section

The study was approved by the Danish Data Protection Agency (journal No. VD-2018-264, with I-Suite no 6504). As this study did not involve contact with patients or an intervention, it was not necessary to obtain permission from the Danish scientific ethical committee.

### 2.1. Study Design

This was a retrospective, 2-year follow-up, register-based cohort study using the following Danish registries: 1. Danish nationwide register of outpatients with COPD (DrCOPD—Danish Register of Chronic Obstructive Pulmonary Disease). From this we gained the following variables: age, gender, forced expiratory volume in 1 s (FEV1), Medical Research Council dyspnea score (MRC), body mass index (BMI), date of out-patient visits, and the date of death; 2. The Danish National Patient Registry (DNPR), which holds information of all admissions at Danish hospitals, registered with International Classification of Disease 10th revision (ICD-10) codes. We extracted the following A or B admission diagnosis: J12–18 (influenza and pneumonia), J20–J22 (other acute lower respiratory infections), and J40–44 (chronic lower respiratory diseases). For the hospitalization-requiring pneumonia outcome, we combined J12–18 with J20–J22. For the AECOPD-Hosp, we only used the codes J40–J44; 3. The Danish National Health Service Prescription Database (DNHSPD), which holds information on all prescriptions for every citizen in Denmark since 2004.

### 2.2. Participants Population

We included all the participants registered with outpatient clinic visits. Study entry was defined as the first outpatient clinic visit for the non-roflumilast group. As far as the roflumilast group was concerned, we included them after the first prescription of roflumilast was received after their first outpatient clinic visit. Exposure to roflumilast was defined as collection of at least one prescription of ATC-code R03DX07 after study entry.

Since roflumilast became available in January 2010 in Denmark, we included participants from 1 January 2010 to 31October 2017.

We excluded all participants who had a malignant cancer diagnosis in the preceding five years, since malignant diagnosis may both have influenced outcomes and the possibility to treat with roflumilast (Table 1). Participants with carcinoma in situ were included (Figure 1).

Additionally, very few participants with unknown vital status were excluded; some participants emigrated from Denmark, and some participants disappeared (Figure 1).

A few participants received all their roflumilast prescriptions before study entry; these participants were allocated to the “non-roflumilast” group since we did not expect the drug to have a prolonged effect after discontinuation.

Missing values for the FEV1 and MRC variables in a minority of participants were handled by carrying the last observation forward. In a few participants, we used the median value (imputation). This strategy was used to avoid selection bias. We had 8011 participants with unknown smoking status. These participants remained in this category. However, we also conducted the main analysis while excluding patients with unknown smoking status.

All comorbidities within five years prior to study entry were entered. The participants had to have at least one hospital contact where the comorbidity was registered as an A diagnosis or B diagnosis in the Danish National Patient Registry (DNPR) to be considered a participant with a comorbidity. They were identified with the ICD10 codes.

In a sensitivity analysis, we included participants who had either roflumilast administration or, in the same calendar period, first administration of theophylline (i.e., active comparator to account for the “new user” effect). Theophylline is also used as a ‘’last resort treatment’’ similar to roflumilast.

### 2.3. Outcomes

Three outcomes were examined: hospitalization-requiring pneumonia, AECOPD-Hosp, and all-cause mortality, respectively, after two years.

### 2.4. Statistics

Baseline characteristics were tested with the chi-square and Wilcoxon test for categorical and continuous variables, respectively. For roflumilast participants, we used the first roflumilast prescription after the first outpatient visit as the baseline. For the control group, we used their first outpatient visit date in DRCOPD. Both groups of participants were then followed for two years. The Cox proportional hazard model was checked for linearity of continues variables, lack of relevant interactions, and proportional hazards and was found to be valid.

### 2.5. Main Analysis: Propensity Score Matching

In order to propensity score-match the participants, we used the Greedy Match from the Mayo Clinic [22]. We matched the two groups on FEV1 (% of predicted), gender (male vs. female), age, ICS (yes vs. no), LAMA/LABA (yes vs. no), number of AECOPD-Hosp one year prior to baseline (≥1 AECOPD-Hosp prior to baseline), MRC (1–5), and smoking status (active vs. former or never vs. unknown status).

We used the unadjusted Cox proportional hazard model to conduct the survival analyses on the matched population. Since mortality was comparable among participants in the roflumilast group and the control group, we did not adjust for mortality during the main outcome analyses (competing risk). We conducted all analyses using SAS 9.4, Cary, NC, USA.

### 2.6. Sensitivity Analysis

Roflumilast vs. The ophylline. We used theophylline as an active comparator drug, which was initiated in the same calendar period as the roflumilast participants had roflumilast initiated. We matched the participants using the same matching variables as before in the Greedy matching algorithm, but now also matched on a new prescription of theophylline vs. new prescription of roflumilast. In total 1123 participants used theophylline in the control group, and we matched 1:1 (propensity score matching). There were 474 participants in the roflumilast group versus 474 participants in the theophylline group (Table 1). We also performed the unadjusted Cox proportional hazard model for the survival analysis for the propensity score-matched roflumilast group to the theophylline control group.

Unmatched population. We used an adjusted cox proportional hazard model for the unmatched population. We adjusted for the same variables as we matched for in the propensity score matching: FEV1, gender, age, ICS, LAMA/LABA, number of AECOPD-Hosp one year prior to baseline, MRC, and smoking status.

Incidence rate (IR).We calculated the number of AECOPD-Hosp one year prior to baseline and one year after baseline for the roflumilast group in order to compare the incidence rate of AECOPD-Hosp before the participants started taking roflumilast and after, in order to use the roflumilast participants as their own control (i.e., cross-over design).

## 3. Results

### 3.1. Patient Characteristics

We identified 45,386 participants with COPD, of whom 594 (1.3%) participants received roflumilast. These roflumilast participants were matched with 2970 non-roflumilast participants. Table 1 shows the demographic and clinical characteristics before and after propensity matching. In the propensity-matched population, there were 1501 males and 1469 females in the non-roflumilast group. In the roflumilast group, there were 297 male participants and 297 female participants. The median age for both groups was 67 years. The number of participants having an AECOPD-Hosp one year prior to baseline in both the roflumilast group and the matched controls was 59%. Overall, the propensity score-matched population and theophylline-matched population was comparable on most baseline characteristics (Table 1). In the unmatched population, the FEV1 median, smoker status, and ≥1 AECOPD was particularly different between the two groups. The difference was also statistically significant (FEV1 median: *p* < 0.0001, smoker status: *p* = 0.0114, and ≥1 AECOPD: *p* < 0.0001).

### 3.2. Main Outcome Analysis

For the propensity-score matched population, the risk of hospitalization-requiring pneumonia was also higher for the roflumilast group after the two year follow-up period (HR 1.5, 1.3 to 1.8 *p* < 0.0001) (Table 2 and Figure 2), the risk of AECOPD-Hosp after two years of follow-up was higher for the roflumilast group (HR 1.6, 1.5 to 1.8 *p* < 0.0001) (Table 2 and Figure 3), and the risk of all-cause mortality (Table 2 and Figure 4) was the same for both groups in the propensity score-matched population (HR 1.0, 95%CI 0.9 to 1.2 *p* = 0.72).

To account for possible misclassification, we conducted the main analysis with the exclusion of the patients with an unknown smoking status and then propensity-matched the two groups. There were 538 participants in the roflumilast group and 2690 participants in the non-roflumilast group. The results still showed an increased risk of hospitalization-requiring pneumonia and AECOPD-Hosp in the roflumilast group (respectively HR = 1.6, 1.4 to 1.9 and *p* < 0.0001 and HR = 1.7, 1.5 to 1.9 *p* < 0.0001), and the signal for all-cause mortality was also unchanged (HR = 0.9, 0.8 to 1.1 *p* = 0.6).

### 3.3. Sensitivity Analysis

For the roflumilast vs. theophylline analysis of the propensity-matched cohort, the risk for hospitalization-requiring pneumonia and AECOPD-Hosp (Table 2, Figure 2 and Figure 3) was higher for the roflumilast group after two years of follow-up (HR 1.5, 1.2 to 1.9; *p* = 0.0005 and HR 1.4, 95% CI 1.2 to 1.6; *p* = 0.0001, respectively). All-cause mortality (Table 2 and Figure 4) did not differ between these two groups (HR 1.2, 95% CI, 0.9 to 1.5, *p* = 0.17).

We also developed an adjusted cox proportional hazard model for the unmatched population while adjusting for the same confounding variables as we used to match in the main analysis. In this analysis, the results regarding hospitalization-requiring pneumonia (Table 2 and Figure 2) and AECOPD-Hosp were largely unchanged as compared to the main analysis, although all-cause mortality seemed slightly higher for the roflumilast group (aHR 1.2, 95% CI 1.0 to1.4; *p* = 0.04) in this secondary analysis (Figure 4).

Finally, we analyzed the incidence rate (IR) of AECOPD-Hosp only for the roflumilast group before and after initiation of roflumilast; the risk was higher in the time after initiation than in the preceding year (Figure 5), *p* < 0.0001 (Mann–Whitney U-test).

All endpoints analyses of 30 days, 90 days, 180 days, and 365 days showed overall an unchanged signal compared to the main analysis. We then combined the endpoint AECOPD with hospitalization-requiring pneumonia and the results pointed overall in the same direction as the separate endpoints (propensity-matched population: HR 1.7, 1.5 to 1.9 *p* < 0.0001; theophylline-matched population: HR 1.3, 95% 1.1 to 1.6; *p* = 0.0002 and unmatched population: aHR 1.7, 95% CI 1.5 to 1.8, *p* < 0.0001). We then did the same analysis for participants having at least two prescriptions of roflumilast for the main outcome to account for the possibility of divergence of effect between “consistent users” and “non-consistent users.” In the roflumilast group, 458 patients out of 594 patients were consistent users (had at least two prescriptions). The results still pointed in the same direction: HR 1.5, 95% CI 1.3 to 1.6, *p* < 0.0001.

## 4. Discussion

### 4.1. Principal Findings

We found that the use of roflumilast for COPD outpatients was associated with an increased risk of hospitalization-requiring pneumonia and hospitalization-requiring acute exacerbations of COPD for a follow-up period of two years. All-cause mortality did not differ between roflumilast patients and their matched controls. These results were robust for analysis with an active comparator (theophylline) and when using patients as their own control: the incidence of hospitalization-requiring acute exacerbations of COPD was higher in the same patients after initiation of roflumilast than before.

### 4.2. Comparison with Other Studies

To our knowledge, the data we report are the first to ever report specifically on hospitalization-requiring pneumonia in roflumilast users. This is of special interest since animal studies have shown that the immune-suppressive effect of roflumilast may have detrimental effects on the host response towards bacterial infections [19], and in being consistent with these, our data show a concerning high risk of pneumonia-hospitalizations in these severely ill, vulnerable COPD patients.

Regarding severe AECOPD, our results point in the same direction (increased risk) which is surprising, considering the trial data that have either showed no effect or reduced risk of this endpoint. However, severe AECOPDs may be caused by either bacteria, or non-bacterial—often viral—inflammation. A possible explanation for this seeming discrepancy could be that the trial populations in general are not at as high of a risk of bacterial infection as our real-life cohort. In fact, patients who have a “bacterial infection-phenotype” have been excluded from some of these trials: in the REACT trial, patients who had lower respiratory tract infections prior to screening and those who used antibiotics or who had bronchiectasis were excluded. Having such exclusion criteria in roflumilast trials may compromise the ability of these trials to correctly estimate the risk of bacterial infections (both as “bacterial AECOPDs” and pneumonias). Additionally, pneumonia outcomes have not been reported in these trials [10,13,23].

Another possible explanation for the discrepancy between our results and the trial results regarding severe AECOPD could be residual bias by indication. However, the consistency of our results through propensity matching, using an active comparator and adjusted analyses, does argue against this as the main explanation. Further, all-cause mortality was equal between roflumilast users and non-users, which argues against pronounced residual bias by indication. In the trials, failure to complete the treatment for the trial period has been observed to be substantially higher among roflumilast-using patients (22% vs. 11%, roflumilast 500 µg vs. placebo) [13], which may also cause a “healthy user effect” thus biasing the results towards larger effect sizes (since non-responders and those with events may discontinue at a higher rate). This may again lead to uncertainty of the true effect of roflumilast for preventing severe exacerbations since it is not entirely clear from the trial publications how drop out patients were handled statistically in regards to severe exacerbations, although it should be acknowledged that drug handling was “good clinical practice” monitored in the trials. Long-term effects for more than one year of follow-up have not been reported in the trials. The majority of the trials, regardless of result, have included less severely ill patients compared with patients in our study, where FEV1 was 34% of predicted for the roflumilast group. We note that in the studies with the most severely ill patients where FEV1 almost resembles the FEV1 of our patients [15], smaller effect sizes were observed [15].

The effect of roflumilast on the inflammatory system in the lungs is not fully understood, but it is known that the acetyl-proline-glycine-proline (AcPGP) pathway takes part in the neutrophilic inflammation in COPD [24,25]. In a 12-week placebo-controlled, randomized study [26], it was demonstrated that the roflumilast treatment decreased the production of acetyl-proline-glycine-proline by >50%. Roflumilast reduces the inflammatory activity by lowering prolyl endopeptidase activity. Even though the effect of roflumilast on neutrophilic inflammation suggests that the drug works on microorganisms, recent studies show that neutrophils have a more complex function: they produce several cytokines and inflammatory factors that play a role on regulating inflammation and the immune system [27,28].

### 4.3. Strengths and Limitations

This study is, to our knowledge, the first observational two-year nationwide registry study conducted to explore the impact of roflumilast on hospitalization-requiring pneumonia and severe (hospitalization-requiring) acute exacerbations of COPD in a real-life outpatient cohort. We had a large sample size with more than 45,000 patients with COPD who met the inclusion criteria. We used Danish national registries with complete data about our study population, including no patient lost to follow-up for the explored outcome measures, and our Nationwide COPD register allowed for the control of important confounders such as FEV1, BMI, and smoking status. Moreover, the COPD diagnosis was made by a specialist in respiratory medicine and verified at each outpatient visit (every 3–12 months). For the main analysis, we propensity score-matched the roflumilast population with the control group, and in a sensitivity analysis we introduced an active comparator, theophylline, also used as a “last resort” treatment. Finally, we did an analysis using the entire patient cohort. The results were markedly unchanged by the analysis method.

Our study also had some limitations. First, although our study population is large, only 594 patients used roflumilast. This might have an impact on power for our different analyses. Second, even though we did our best to match in the propensity score-matched cohort and to introduce an active comparator (theophylline) to account for this, by adjusting for known confounders and severity of the disease, we cannot exclude some residual confounding factors or bias by indication. Furthermore, patients with COPD who used roflumilast were included from the first roflumilast prescription after an outpatient visit. These patients were then automatically followed in a longer period in the DrCOPD. The reason for the longer follow-up time in a lung specialist clinic might be an illustration of them being sicker. Finally, it is challenging to monitor the patients’ adherence to the drug. We cannot be sure that the patients in the roflumilast group took the drug since it is a retrospective registry study, and this is an inborn error of the design. However, we did two analyses: in the first analysis we defined consistent users as patients with at least one prescription, and in the second analysis we defined it as patients with at least two prescriptions. Both analyses pointed in the same direction. We cannot exclude this as a possible explanation for an ineffectiveness of the drug, but we cannot explain an increased risk of pneumonia and severe AECOPD when compared to non-users.

## 5. Conclusions

In this nationwide study with complete follow-up, we found an increased risk of hospitalization-requiring pneumonia and hospitalizations due to acute COPD exacerbation, which is biologically plausible (immune-suppression leading to incompetence to handle bacterial infections). This is concerning since roflumilast is considered a last resort treatment in the most severely ill COPD patients, who are often at high risk of bacterial infections.

Trial data cannot enlighten this area since “bacterial infection phenotype” patients may have been excluded, and since bacterial infection complications are not reported consistently in the trials. Additionally, there is a highly skewed drop-out of the trials which may tend to overestimate the positive effects of the drug.

Trials are urgently needed to determine the risk of bacterial infectious complications to roflumilast in COPD patients at risk of bacterial infections like pneumonias. Patients receiving this drug in real clinical life are probably at a higher risk of such complications than the trial populations, and we recommend very cautious use of this drug in COPD patients at risk of bacterial infections.

## Figures and Tables

**Figure 1 jcm-09-01442-f001:**
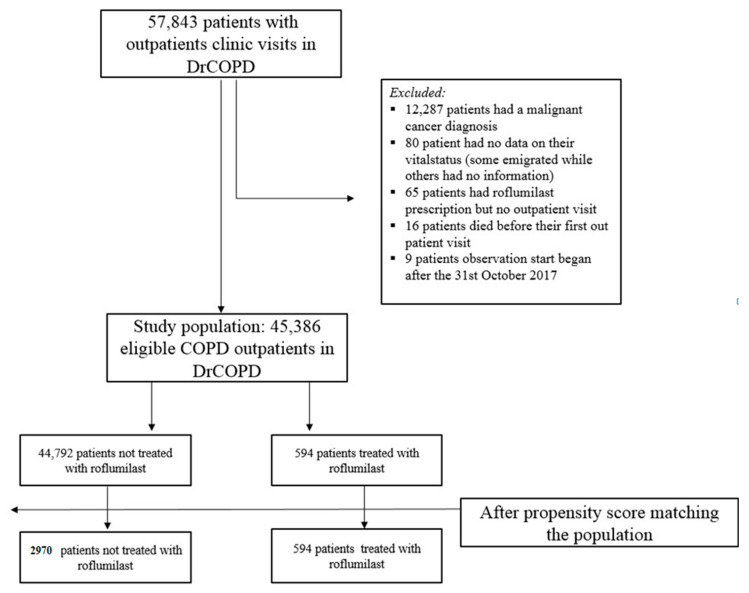
Participants population.

**Figure 2 jcm-09-01442-f002:**
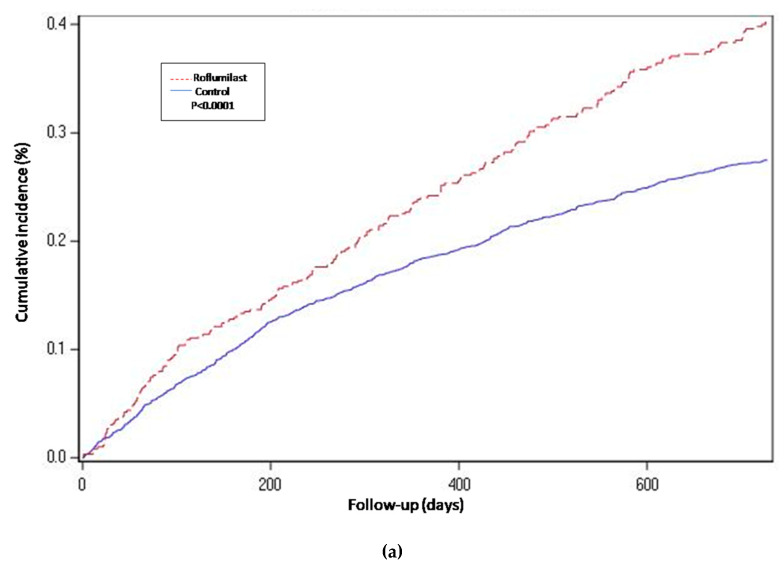
Years of follow-up. Hospitalization-requiring pneumonia. Cumulative incidence curves for the three compared groups. (**a**) Propensity-matched population (Roflumilast vs. Control): Unadjusted Cox proportional hazard model (matched on FEV1, MRC, age, gender, smoke status, number of AECOPD-Hosp 1 year prior to baseline, ICS prescription and LAMA/LABA prescription). (**b**) Roflumilast versus Theophylline: unadjusted Cox proportional hazard mode (Propensity score matched on FEV1, MRC, age, gender, smoke status, number of AECOPD-Hosp 1 year prior to baseline, ICS prescription, LAMA/LABA prescription and calendar year). (**c**) Unmatched population (Roflumilast vs. Control): adjusted Cox proportional hazard model (adjusted for FEV1, MRC, age, gender, smoke status, number of AECOPD-Hosp 1 year prior to baseline, ICS prescription and LAMA/LABA prescription). Follow-up period of two years.

**Figure 3 jcm-09-01442-f003:**
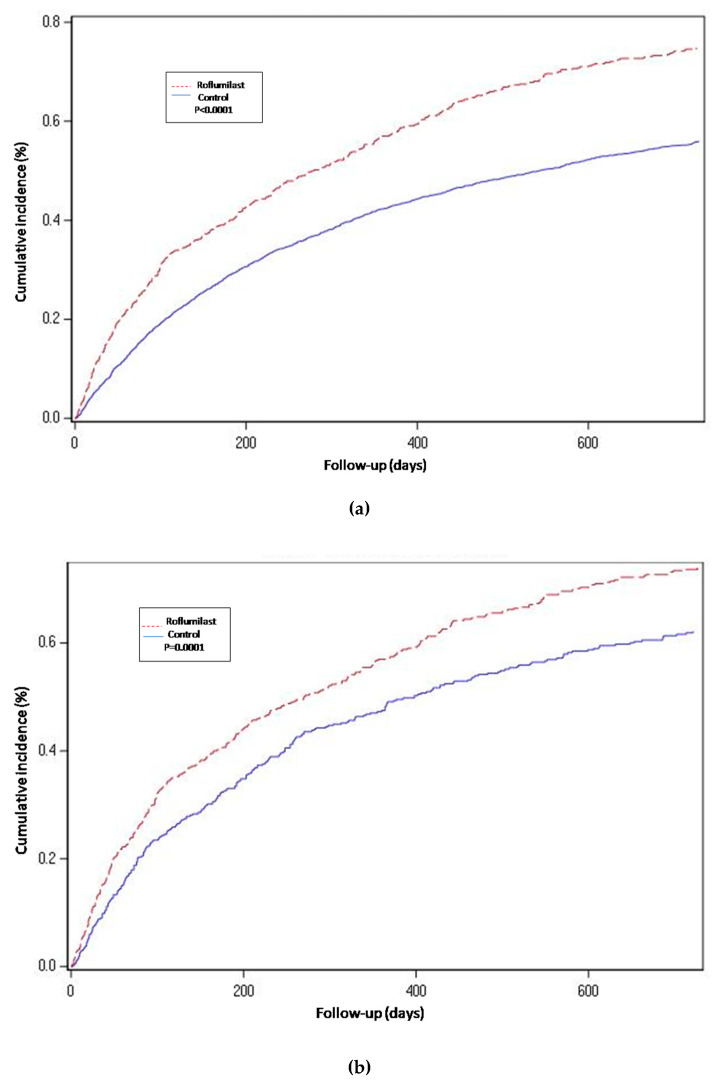
Hospital-requiring acute COPD exacerbation. Cumulative incidence for the three compared groups. (**a**) Propensity-matched population (Roflumilast vs. Control): Unadjusted Cox proportional hazard model (matched on FEV1, MRC, age, gender, smoke status, number of AECOPD-Hosp 1 year prior to baseline, ICS prescription and LAMA/LABA prescription). (**b**) Roflumilast versus Theophylline: unadjusted Cox proportional hazard mode (Propensity score matched on FEV1, MRC, age, gender, smoke status, number of AECOPD-Hosp 1 year prior to baseline, ICS prescription, LAMA/LABA prescription and calendar year). (**c**) Unmatched population (Roflumilast vs. Control): adjusted Cox proportional hazard model (adjusted for FEV1, MRC, age, gender, smoke status, number of AECOPD-Hosp 1 year prior to baseline, ICS prescription and LAMA/LABA prescription). Follow-up period of two years.

**Figure 4 jcm-09-01442-f004:**
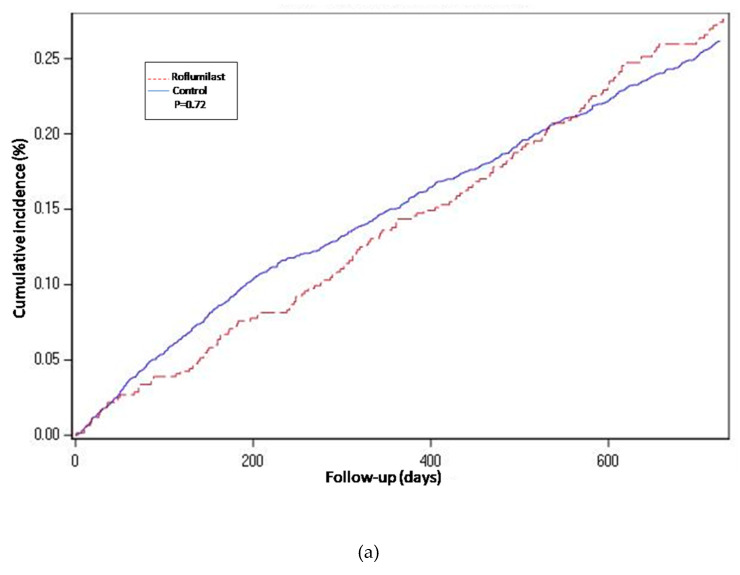
Cumulative incidence curves for all-cause mortality. (**a**) Propensity-matched population (Roflumilast vs. Control). (**b**) Roflumilast versus Theophylline. (**c**) Unmatched population (Roflumilast vs. Control).

**Figure 5 jcm-09-01442-f005:**
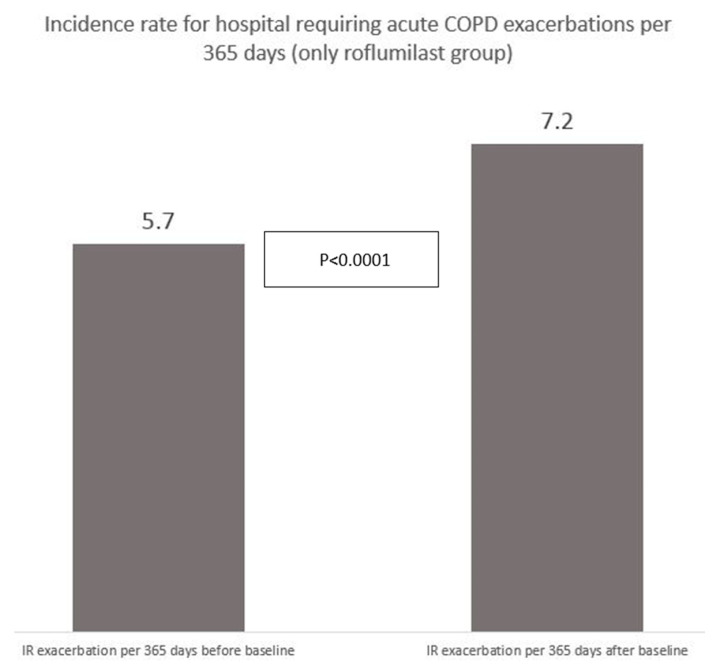
Incidence rate of hospitalization-requiring acute exacerbations for 594 participants receiving roflumilast before and after starting up with roflumilast (AECOPD/365 days).

**Table 1 jcm-09-01442-t001:** The demographic and clinical characteristics for the entire population, the propensity-matched population and the theophylline matched population.

*N* (Number of Participants)	Entire COPD Cohort *n* = 45,386		Propensity-Matched Cohort *n* = 3564		Propensity-Matched Cohort + Theophylline-Matched *n* = 948	
	Non-Roflumilast Cohort	Roflumilast Cohort	Non-Roflumilast Cohort	Roflumilast Cohort	Theophylline Cohort	Roflumilast Cohort
Characteristics	*n* = 44,792	*n* = 594	*n* = 2970	*n* = 594	*n* = 474	*n* = 474
Demographics						
Age, median (IQR)	70.5 (62.4–77.9)	67 (60.5–72.4)	67.3 (59.6–74.7)	67 (60.5–72.4)	67.5 (61.4–74.6)	67.6 (61.7–73.3)
Age <63 (*n* (%))	11,849 (26.4)	195 (32.8)	1024 (34.5)	195 (32.8)	142 (30)	137 (28.9)
63–70 (*n* (%))	9733 (21.7)	188 (31.7)	757 (25.5)	188 (31.7)	134 (28.27)	145 (30.59)
71–77 (*n* (%))	10,718 (23.9)	137 (23.1)	646 (21.8)	137 (23.1)	110 (23.21)	121 (25.5)
>78 (*n* (%))	12,492 (27.9)	74 (12.5)	543 (18.3)	74 (12.5)	88 (18.6)	71 (15)
Male (%)	22,236 (49.6)	297 (50)	1501 (50.5)	297 (50)	233 (50.7)	230 (50)
Female (%)	22,556 (50.4)	297 (50)	1469 (49.5)	297 (50)	241 (49.3)	244 (50)
FEV1(%) median (IQR)	49 (36–63)	34 (26–44)	33 (24–46)	34 (26–44)	36 (25–49)	35 (27–45)
MRC (1–5) median (IQR)	3 (2–4)	4 (3–4)	4 (3–4)	4 (3–4)	3 (3–4)	4 (3–4)
BMI (kg/m^2^) median (IQR)	25 (21–29)	25 (22–28)	24 (20–28)	25 (22–28)	25 (21–29)	25 (22–28)
Smoker (%)	13,771 (30.3)	154 (25.9)	876 (29.5)	154 (25.9)	132 (27.9)	125 (26.4)
Ex–smoker/non-smoker (%)	23,066 (50.8)	384 (64.9)	1857 (62.5)	384 (64.65)	295 (62.24)	302 (63.71)
Unknown smoker status (%)	7955 (17.5)	56 (9.4)	237 (9)	56 (9.4)	47 (9.9)	47 (9.9)
LAMA/LABA (%)	37,070 (82.8)	586 (98.7)	2928 (98.6)	586 (98.7)	464 (97.9)	466 (98.3)
ICS (%)	31,844 (71.1)	571 (96.1)	2856 (96.2)	571 (96.1)	450 (94.9)	453 (95.6)
≥1 AECOPD-Hosp (%)	21,361 (47.7)	354 (59.6)	1748 (58.9)	354 (59.6)	290 (61.2)	282 (59.5)
Comorbidities, *n* (%)						
Acute myocardial infarction	2188 (4.9)	18 (3)	135 (4.6)	18 (3)	21 (4.4)	14 (3)
Atrial fibrillation	5886 (13.1)	51 (8.6)	325 (10.9)	51 (8.6)	47 (9.9)	47 (9.9)
Hypertension	11,089 (24.8)	124 (20.9)	643 (21.7)	124 (20.9)	109 (23)	107 (22.6)
Chronic Renal failure	1738 (3.9)	6 (1)	82 (2.8)	6 (1)	10 (2.1)	6 (1.3)
Asthma	3670 (8.2)	74 (12.5)	298 (10)	74 (12.5)	96 (20.3)	62 (13.1)
Depression	1331 (3)	19 (3.2)	82 (2.8)	19 (3.2)	17 (3.6)	14 (3)
Diabetes mellitus	4564 (10.2)	63 (10.6)	252 (8.5)	63 (10.6)	47 (9.9)	56 (11.8)
Cerebrovascular events	2708 (6.0)	21 (3.5)	147 (5)	21 (3.5)	26 (5.5)	19 (4)
Heart failure	5426 (12.1)	57 (9.6)	314 (10.6)	57 (9.6)	48 (10.1)	50 (10.6)
Peripheral arterial disease	3463 (7.7)	31 (5.2)	173 (5.8)	31 (5.2)	24 (5.1)	28 (5.9)

COPD; Chronic Obstructive Pulmonary Disease, MRC; Medical Research Council Dyspnea Scale, BMI; Body Mass Index (kg/m^2^), FEV1; Forced Expiratory Volume in the first second, LAMA/LABA; long-acting muscarinic antagonist, LABA; long-acting beta2 agonist, ICS; inhalation corticosteroids, AECOPD-Hosp; hospitalization requiring acute COPD exacerbation.

**Table 2 jcm-09-01442-t002:** Risk of hospitalization-requiring acute exacerbations after initiation of roflumilast therapy.

		Primary Outcome	Secondary Outcome		*N* Non-Roflumilast	*N* Roflumilast
		AECOPD- Hospitalisation	All-Cause Mortality	Hospitalisation-Requiring Pneumonia		
Propensity-matched population	HR (95% CI) *p*-value	1.6 (1.5 to 1.8) < 0.0001	1.0 (0.9 to 1.2) 0.72	1.5 (1.3 to 1.8) < 0.0001	2970	594
Roflumilast vs. Theophylline		1.4 (1.2 to 1.6) 0.0001	1.2 (0.9 to 1.5) 0.17	1.5 (1.2 to 1.9) 0.0005	474	474
Unmatched population	Adjusted HR (95% CI) *p*-value	1.7 (1.5 to 1.9) < 0.0001	1.2 (1.0 to 1.4) 0.04	1.7 (1.4 to 1.9) < 0.0001	44,792	594

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
