# Peer review of "Roflumilast in Severely Ill Patients with Chronic Obstructive Pulmonary Disease with Frequent Exacerbations: Risk of Pneumonia Hospitalization and Severe Exacerbations"

_jcm, 2020, doi:10.3390/jcm9051442_

Round 1
Reviewer 1 Report
This is a real-life retrospective study, 2-ys follow-up, Danish register-base cohort study on COPD patients with frequent exacerbations. The number of subjetcs in the study population (45,386 outpatients with COPD, including 549 roflumilats cohort) is considerable. The aim of study is very interesting. The authors wanted to determine if roflumilast (an oral selective inibitor of PDE4 with anti-inflammatory action on treatment of COPD with frequent exacerbations and post-bronchodilatory FEV1 less than 50% preducted) is associated with a lower risk of severe COPD exacerbations or hospitalisation-requiring pneumonia or all-cause mortality in COPD outpatients during 2-ys follow-up period. The statistical analysis is robust. The authors performed the analysis in the full cohort and also using a control group of COPD patient in theophylline treatment (using patients as their own control, matched 1:1 with a COPD patient in roflumilast treatment). The figures are well presented, although numbering can be confusing. The results showed that the use of roflumilast for COPD outpatients was associated with an increased risk of hospitalisation-requiring acute exacerbations of COPD and hospitalisation-requiring pnuemonia. These results were highlighted both in the full cohort and also in the control group in theophylline therapy.
Questions
1.The results of the study are not in agreement with other results from RCT studies: Martinez FJ et al. (Lancet 2015) showed in moderate-to-severe COPD exacerbations (- 13·2% in the roflumilast group than in the placebo group) and in severe exacerbations (J Respir Crit Care Med 2018) a significant benefit of roflumilast in decreasing exacerbations (- 13.2% and -16.1% roflumilast vs placebo respectively) (references 10 and 21). These are studies characterized by relevant numbers of severe COPD patients, and therapeutic adherence is monitored and is generally high. In two real-life studies on severe COPD exacerbations, the Spanish study on 55 patients, including 28 subjects that completed 1-year of roflumilast therapy (Munoz-Esquerre M et al. Pulmonary Pharmacology & Therapeutics 2015;30:16-21) and the Turkish study on 83 COPD patients in 6-months roflumilast treatment (Cilli A et al. J Thorac D 2019;(11)4:1100-1105), reported a decrease of COPD exacerbations and hospitalisations with addition of roflumilast. In these studies data about the roflumilast adherence are reported in the text and have been complete. These are undoubtedly studies conducted on a limited number of patients but still they are coming from clinical reality.
At 300 row p. 10 “…for participants having at least two prescriptions of roflumilast…”consistent user” and “non-consistent users”… Only in this point there are some indications on roflumilast adherence. Has the roflumilast adherence been monitored? and if so in what way? How many were the non-consistent users? These aspects should be better specified in the text.
Is it possible that the data of ineffectiveness in severe exacerbations (as better classified in the methods) by roflumilast depends on an incomplete adherence to roflumilast therapy?
- Table A1 p. 13. FEV1 median, smoker (%), ex-smokers (%), severe exacerbation (≥1 AECOPD-Hoso%) seem different in the two groups (non-roflumilast cohort and roflumilast cohort). Is there a significant statistical difference between the two groups? The difference or not between the basal data of the groups should be better specified in the text or in the table.
- To avoid confusion, since all the tables and figures are in appendix A (so the authors reported the letters A at the beginning of any table and figure), it could be better to indicate in the legends the single figures with the lower case letters of the alphabet: a, b, c, d.
- Figure A4 D p. 17 The IR for hospital-requiring acute COPD exacerbations per 365 days in roflumilast group was higher after starting up with roflumilast. Is this increase statistically significant or not? It should be better specified in the text and figure.
Author Response
03rd May 2020
Assistant Editor
Ms. Milesa Blagojevic,
Journal of Clinical Medicine
Dear editors, dear reviewers,
First and foremost, we would like to thank you for taking the time for reading our manuscript (JCM Manuscript ID 754033) as an original research article in Journal of Clinical Medicine, and furthermore for giving us the possibility of revising the manuscript. Thank you for your valuable comments. We have below answered your questions point by point. The reviewer comments are listed C1, C2, C3 for each reviewer, and the responses are listed correspondingly, R_C1, R_C2 etc.
Reviewer 1:
C1
This is a real-life retrospective study, 2-ys follow-up, Danish register-base cohort study on COPD patients with frequent exacerbations. The number of subjetcs in the study population (45,386 outpatients with COPD, including 549 roflumilast cohort) is considerable. The aim of study is very interesting. The authors wanted to determine if roflumilast (an oral selective inibitor of PDE4 with anti-inflammatory action on treatment of COPD with frequent exacerbations and post-bronchodilatory FEV1 less than 50% predicted) is associated with a lower risk of severe COPD exacerbations or hospitalisation-requiring pneumonia or all-cause mortality in COPD outpatients during 2-ys follow-up period. The statistical analysis is robust. The authors performed the analysis in the full cohort and also using a control group of COPD patient in theophylline treatment (using patients as their own control, matched 1:1 with a COPD patient in roflumilast treatment). The figures are well presented, although numbering can be confusing. The results showed that the use of roflumilast for COPD outpatients was associated with an increased risk of hospitalisation-requiring acute exacerbations of COPD and hospitalisation-requiring pnuemonia. These results were highlighted both in the full cohort and also in the control group in theophylline therapy.
R_C1:
Thank you for this
C2.
The results of the study are not in agreement with other results from RCT studies: Martinez FJ et al. (Lancet 2015) showed in moderate-to-severe COPD exacerbations (- 13·2% in the roflumilast group than in the placebo group) and in severe exacerbations (J Respir Crit Care Med 2018) a significant benefit of roflumilast in decreasing exacerbations (- 13.2% and -16.1% roflumilast vs placebo respectively) (references 10 and 21). These are studies characterized by relevant numbers of severe COPD patients, and therapeutic adherence is monitored and is generally high. In two real-life studies on severe COPD exacerbations, the Spanish study on 55 patients, including 28 subjects that completed 1-year of roflumilast therapy (Munoz-Esquerre M et al. Pulmonary Pharmacology & Therapeutics 2015;30:16-21) and the Turkish study on 83 COPD patients in 6-months roflumilast treatment (Cilli A et al. J Thorac D 2019;(11)4:1100-1105), reported a decrease of COPD exacerbations and hospitalisations with addition of roflumilast. In these studies data about the roflumilast adherence are reported in the text and have been complete. These are undoubtedly studies conducted on a limited number of patients but still they are coming from clinical reality.
R_C2. We absolutely agree with your comments. Previous RCT have shown a moderate effect of roflumilast on acute exacerbations, which is in disagreement with our findings. It is correct that in Martinez et al trial (10), the frequency of moderate or severe exacerbations was 13.2% numerically lower in the roflumilast group versus placebo. Opposite, in the substudy from the REACT trial, no effect was found on severe AECOPD´s. However, this result was only borderline significant, and may be driven mainly by moderate exacerbations.
However, the drug may actually be beneficial in patients who primarily suffer from non-bacterial exacerbations, but deleterious in patients who primarily suffer from bacterial exacerbations and pneumonias. We followed up on this and found experimental data to support that roflumilast may, in fact, be detrimental, in bacterial infections. Kasetty et al found that rodents infected with Gram negative bacterium Pseudomonas aeruginosa who were medicated with roflumilast had higher mortality, fewer neutrophils in BAL-fluid, but higher bacterial counts in BAL fluid. Roflumilast was given in different doses, and interestingly, there was a clear dose-response relationship in the findings (higher roflumilast doses led to higher bacterial counts, higher mortality and lower neutrophil counts in BAL-fluid).
Additionally, in several of the RCT´s patients with a bacterial exacerbation phenotype have been excluded, e.g. the patients who had recently had a LRTI and those with bronciectasies, were excluded. Thus, we think the data so far on roflumilast from randomized trials cannot be properly interpreted yet regarding the specific risk in patients with primarily bacterial exacerbations, since information on how many AECOPD´s were associated to bacterial and viral infections, is not available. Since viral causes are more frequent, but bacterial causes more fatal, it is very possible that data from the effect of roflumilast on viral AECOPD´s (beneficial) are driving the results (“Moderate-severe AECOPD” is by far the most common outcome reported as primary outcome in these trials), but at the same time roflumilast may be causing an increase in the risk of bacterial AECOPD´s and possibly death.
This has now been added to the introduction and the discussion:
This part is now elucidated in the introduction:
“Of special concern, animal experiments have shown deleterious effects of roflumilast in bacterial lung infectious with increased mortality and bacterial loads(19), and likewise concerning, roflumilast has been documented to reduce the release of chemokine cysteine‐cysteine ligand (CCL)2, CCL3, CCL4, chemokine cysteine‐X‐cystein ligand (CXCL) 10 and TNF‐α, which may severely compromise the host defense towards bacterial infections(20, 21). However, specific results regarding the risk of roflumilast on hospitalization-requiring pneumonias have not been reported in trial data, and data on severe (hospitalization-requiring) COPD exacerbations have been sparse and diverging (10)]. Additionally, to our knowledge, long-term follow-up for outcomes in patients with severe COPD using roflumilast have not been investigated in larger real-life cohorts.”
And in the discussion, this is commented:
“To our knowledge, the data, we report, are the first ever human data to report specifically on hospitalization-requiring pneumonia in roflumilast users. This is of special interest, since animal studies have shown that the immune-suppressive effect of roflumilast, may have detrimental effects on the host response towards bacterial infections(19), and in consistence with these, our data show a concerning high risk of pneumonia-hospitalizations in these severely ill vulnerable COPD patients.
Regarding severe AECOPD, our results point in the same direction (increased risk) which is surprising, considering the trial data, that have either showed no effect or reduced risk of this endpoints. However, severe AECOPD´s may be caused by either bacteria, or non-bacterial – often viral – inflammation. A possible explanation for this seeming discrepancy, could be that the trial populations in general are not in as high risk of bacterial infection as our real-life cohort, and in fact, patients who have a “bacterial infection-phenotype” have been excluded from some of these trials: in the REACT trial, patients who had lower respiratory tract infections prior to screening, and those who used antibiotics or who had bronchiectasis, were excluded. Having such exclusion criteria in roflumilast trials may compromise the ability of these trials in correctly estimating the risk of bacterial infections (both as “bacterial AECOPD´s” and pneumonias). Additionally, pneumonia outcomes have not been reported in the trials.”
We can see that the adherence has been complete in the previous studies. We have added the following sentence in the manuscript:
“This may again lead to uncertainty of the true effect of roflumilast for preventing severe exacerbations, since it is not entirely clear from the trial publications how drop out patients were handled statistically in regard to severe exacerbations, although it should be acknowledged that drug handling was Good-clinical practice monitored in the trials. ”
C3. At 300 row p. 10 “…for participants having at least two prescriptions of roflumilast…”consistent user” and “non-consistent users”… Only in this point there are some indications on roflumilast adherence. Has the roflumilast adherence been monitored? and if so in what way? How many were the non-consistent users? These aspects should be better specified in the text.
R_C3. Thanks for this comment. As you also point out, we know it is not possible to assure 100% adherence to the drug since it is a retrospective study. Nevertheless, we looked at the roflumilast prescription. We managed to do two analyses. In the first analysis, we assumed that if a patient had one roflumilast prescription then the patient belonged to the roflumilast group. However, one prescription of roflumilast is not consistent with being a consistent user of the drug. So, we made another analysis where we defined a consistent roflumilast user as a patient with at least two prescriptions. The results still pointed in the same direction as the first analysis (no signal change). In the roflumilast group 458 patients out of 594 patients were consistent users (had at least 2 prescriptions). We have now made this clearer in the manuscript:
“In the roflumilast group 458 patients out of 594 patients were consistent users (had at least 2 prescriptions).“.
C4. Is it possible that the data of ineffectiveness in severe exacerbations (as better classified in the methods) by roflumilast depends on an incomplete adherence to roflumilast therapy?
R_C4. Thanks for this valuable comment. We know that frequent development of adverse events and consequent low adherence are major barriers in the wide use of roflumilast. However, we must expect these side effects to occur after redeeming of the first prescription. A several our patients also redeemed the number 2 prescription (458 out of 594). Therefore, we consider it unlikely that this can explain our findings. Additionally, our findings were not neutral (which should have been expected if non-adherence was the explanation). Our results indicate harm from the drug, which should not be expected if patients were merely not taking the drug.
We have added the following to the limitation section of the discussion:
“Finally, it is challenging to monitor the patients’ adherence to the drug. We cannot be sure that the patients in the roflumilast group took the drug since it is a retrospective registry study, and this is an inborn error of the design. However, we did two analyses: first analysis we defined consistent users as patients with at least one prescription and in the second analysis we defined it as patients with at least two prescriptions. Both analyses pointed in the same direction. We cannot exclude this as a possible explanation for an ineffectiveness of the drug, but we non-adherence cannot explain an increased risk of pneumonia and severe AECOPD, when compared to non-users “
C5.
Table A1 p. 13. FEV1 median, smoker (%), ex-smokers (%), severe exacerbation (≥1 AECOPD-Hoso%) seem different in the two groups (non-roflumilast cohort and roflumilast cohort). Is there a significant statistical difference between the two groups? The difference or not between the basal data of the groups should be better specified in the text or in the table.
R_C5
We performed Wilcoxon test for FEV1 since it is a continuous value. P value was <0.0001. Therefore, there was a statistically significant difference between the groups. We performed Chi square test for the smoking status and >=AECOPD. The result was 0.0114 and <0.0001 respectively, which means that there was a statistical significant difference between the two groups. We will add this information to the result section. In our statisctical analysis, we have matched for these variables. We have now added this information in the manuscript:
“In the unmatched population the FEV1 median, smoker status and >=1 AECOPD is particularly different between the two the groups. The difference is also statistically significant (FEV1 median: p<0.0001, smoker status: p=0.0114, and >=1 AECOPD: p<0.0001)”.
Additionally, the main analysis was not the unmatched cohort, but rather the propensity matched cohort, in which the baseline characteristics are more evenly distributed.
C6.
To avoid confusion, since all the tables and figures are in appendix A (so the authors reported the letters A at the beginning of any table and figure), it could be better to indicate in the legends the single figures with the lower case letters of the alphabet: a, b, c, d.
R_C6.
Thank you for your feedback on the figures. We will now indicate in the legends the single figures with the lower-case letters of the alphabet. We apologize that we had placed all the figures in the appendix – they are now placed in the manuscript and named Figure 1, Figure 2, etc.
C7.
Figure A4 D p. 17 The IR for hospital-requiring acute COPD exacerbations per 365 days in roflumilast group was higher after starting up with roflumilast. Is this increase statistically significant or not? It should be better specified in the text and figure.
R_C7. We performed Mann-whitney U test: p<0.0001. Thank you for pointing this out. The information is added in the following way:
“Finally, we analyzed the incidence rate (IR) of AECOPD-Hosp only for the roflumilast group before and after initiation of roflumilast; the risk was higher in the time after initiation than in the preceding year (Figure A5), p<0.0001 (Mann-Whitney-U-test).“
Reviewer 2 Report
In this study authors found that patients with COPD who received roflumilast had an increased 2-year risk of admission with AECOPD-Hosp and hospitalization-requiring pneumonia compared to controls. All endpoints analyses of 30 days, 90 days, 180 days and 365 days showed overall an unchanged result compared to the main analysis. When adding an active comparator (theophylline) as a matching variable, AECOPD-Hosp and hospitalization-requiring pneumonia rate remained higher in the roflumilast group.
At this regard we have to point out that this was a retrospective, 2-year follow-up, cohort study. Patients’ data have been retrospectively collected from three Danish registries (the Danish nationwide register of outpatients with COPD, the Danish National Patient Registry and the Danish National Health Service Prescription Database). This could have implied a bias in roflumilast indication and consequently in the selection of the study population. Nevertheless only 357 patients used roflumilast versus 2,970 not treated patients in the control group and this may have represented another bias in the results of the applied statistic analysis.
In my opinion, despite this study have the merit to have explored roflumilast efficacy in a larger follow-up period (2 years), further longitudinal homogeneous controlled study are needed before a precise conclusion can be drawn.
In the discussion section authors proposed that roflumilast may be more beneficial in moderating the viral infection host response than the bacterial infection host response. This is an interesting hypothesis that should be more emphasized a could represent a valid idea for further studies.
Author Response
Reviewer 2:
In this study authors found that patients with COPD who received roflumilast had an increased 2-year risk of admission with AECOPD-Hosp and hospitalization-requiring pneumonia compared to controls. All endpoints analyses of 30 days, 90 days, 180 days and 365 days showed overall an unchanged result compared to the main analysis. When adding an active comparator (theophylline) as a matching variable, AECOPD-Hosp and hospitalization-requiring pneumonia rate remained higher in the roflumilast group.
C1.
At this regard we have to point out that this was a retrospective, 2-year follow-up, cohort study. Patients’ data have been retrospectively collected from three Danish registries (the Danish nationwide register of outpatients with COPD, the Danish National Patient Registry and the Danish National Health Service Prescription Database). This could have implied a bias in roflumilast indication and consequently in the selection of the study population. Nevertheless only 357 patients used roflumilast versus 2,970 not treated patients in the control group and this may have represented another bias in the results of the applied statistic analysis.
R_C1.
Thank you for this comment. Small correction. In the propensity matched cohort there were 594 roflumilast users (not 357) the matched controls were correctly 2,970.
We agree that our results could in part be explained by bias by roflumilast indication. We have now expanded this part of the limitations section in the discussion:
“Our study also had some limitations. First, although our study population is large, only 594 patients used roflumilast. This might have an impact on power for our different analyses. Second, even though we did our best to match in the propensity score matched cohort, and further to introduce an active comparator (theophylline) to account for this, and by adjusting for known confounders and severity of the disease, we cannot exclude some residual confounding or bias, bias by indication.”
C2.
In my opinion, despite this study have the merit to have explored roflumilast efficacy in a larger follow-up period (2 years), further longitudinal homogeneous controlled study are needed before a precise conclusion can be drawn.
R_C2.
We strongly agree that further longitudinal homogenous controlled studies are needed in order to draw a more precise conclusion. We have now corrected our conclusion. It is now displayed in the following way in the manuscript:
“In this nationwide study with complete follow-up, we found an increased risk of hospitalization-requiring pneumonia and hospitalizations due to acute COPD exacerbation, which is biologically plausible (immune-suppression leading to incompetence to handle bacterial infections). This is concerning since roflumilast is considered a last resort treatment in the most severely ill COPD patients, who are, in real life, often at high risk of bacterial infections.
Trial data cannot enlighten this area since “bacterial infection phenotype” patients may have been excluded, and since bacterial infection complications are not reported consistently in the trials. Additionally, there is a highly skewed drop out of the trials, which may tend to overestimate the positive effects of the drug.
Trials are urgently needed to determine the risk of bacterial infectious complications to roflumilast in COPD patients at risk of bacterial infections like pneumonias. Patients receiving this drug in real clinical life are probably at a higher risk of such complications than the trial populations, and we recommend very cautious use of this drug in COPD patients at risk of bacterial infections.”
C3.
In the discussion section authors proposed that roflumilast may be more beneficial in moderating the viral infection host response than the bacterial infection host response. This is an interesting hypothesis that should be more emphasized a could represent a valid idea for further studies.
R_C3.
Thank you for your valuable comment. We think that the roflumilast could have a better effect for viral infections than for bacterial infections. We have extensively expanded this point in the introduction:
“Of special concern, animal experiments have shown deleterious effects of roflumilast in bacterial lung infectious with increased mortality and bacterial loads(19), and likewise concerning, roflumilast has been documented to reduce the release of chemokine cysteine‐cysteine ligand (CCL)2, CCL3, CCL4, chemokine cysteine‐X‐cystein ligand (CXCL) 10 and TNF‐α, which may severely compromise the host defense towards bacterial infections(20, 21). However, specific results regarding the risk of roflumilast on hospitalization-requiring pneumonias have not been reported in trial data, and data on severe (hospitalization-requiring) COPD exacerbations have been sparse and diverging (10)].”
And in the discussion:
“To our knowledge, the data, we report, are the first ever human data to report specifically on hospitalization-requiring pneumonia in roflumilast users. This is of special interest, since animal studies have shown that the immune-suppressive effect of roflumilast, may have detrimental effects on the host response towards bacterial infections(19), and in consistence with these, our data show a concerning high risk of pneumonia-hospitalizations in these severely ill vulnerable COPD patients.
Regarding severe AECOPD, our results point in the same direction (increased risk) which is surprising, considering the trial data, that have either showed no effect or reduced risk of this endpoints. However, severe AECOPD´s may be caused by either bacteria, or non-bacterial – often viral – inflammation. A possible explanation for this seeming discrepancy, could be that the trial populations in general are not in as high risk of bacterial infection as our real-life cohort, and in fact, patients who have a “bacterial infection-phenotype” have been excluded from some of these trials: in the REACT trial, patients who had lower respiratory tract infections prior to screening, and those who used antibiotics or who had bronchiectasis, were excluded. Having such exclusion criteria in roflumilast trials may compromise the ability of these trials in correctly estimating the risk of bacterial infections (both as “bacterial AECOPD´s” and pneumonias). Additionally, pneumonia outcomes have not been reported in the trials. [10, 13, (24)]
Round 2
Reviewer 1 Report
The authors made the requested corrections , which are consistent with the previous comments of the first revision and reasonable overall.
It should be noted that in the paragraph of the statistical analysis the new numbering of the figures has not been inserted
Author Response
Reviewer 1:
C1.
The authors made the requested corrections , which are consistent with the previous comments of the first revision and reasonable overall.
It should be noted that in the paragraph of the statistical analysis the new numbering of the figures has not been inserted
R_C1.
Thank you for your quick response and kind comment. We have inserted the new numbering of the figures in the paragragh of the statistical analysis.
“The number of participants having an AECOPD-Hosp 1-year prior to baseline in both the roflumilast group and the matched controls was 59% Overall, the propensity score matched population and theophylline matched population was comparable on most baseline characteristics (table 1). “(line 168-171).